# Extracts from Tartary Buckwheat Sprouts Restricts Oxidative Injury Induced by Hydrogen Peroxide in HepG2 by Upregulating the Redox System

**DOI:** 10.3390/foods13233726

**Published:** 2024-11-21

**Authors:** Xiaoping Li, Yuwei Zhang, Wen Zhao, Tian Ren, Xiaolong Wang, Xinzhong Hu

**Affiliations:** 1College of Food Engineering and Nutritional Science, Shaanxi Normal University, Xi’an 710119, China; zhaowen091110@163.com (W.Z.); rentian@snnu.edu.cn (T.R.); wangxiaolong23@126.com (X.W.); hxinzhong@snnu.edu.cn (X.H.); 2School of Food Science and Engineering, South China University of Technology, Guangzhou 510640, China; katezhang@126.com

**Keywords:** oxidative stress, phenolic compounds, tartary buckwheat sprouts, HepG2 cells, nucleus factor E2 related factor 2

## Abstract

Oxidative stress, which results from an overproduction of reactive oxygen species (ROS), can cause damage that may contribute to a range of metabolic disorders. Antioxidants are considered to upregulate the activity of antioxidant enzymes, which are crucial for eliminating excess ROS and safeguarding the body against oxidative stress-induced damage. In the present study, the effect of polyphenol extracts from tartary buckwheat sprouts (TBSE) on the redox system of HepG2-cell-induced oxidative injury by hydrogen peroxide were investigated for evaluating the protective effect and mechanism of tartary buckwheat sprouts (TBS). The results revealed that TBSE that had sprouted for a period of 10 days possessed six predominant phenolic compounds, ranked from the most abundant to the least: chlorogenic acid, syringic acid, caffeic acid, rutin, ferulic acid, and quercetin. TBSE could successfully inhibit H_2_O_2_-induced ROS overproduction, restore and balance the mitochondrial membrane potential, while also significantly increasing cellular antioxidant activity (CAA) and the expression of protective enzymes such as SOD, CAT, and GST. More interestingly, treating HepG2 cells with TBSE triggered the translocation of Nrf2 to the nucleus, accompanied by a negative feedback mechanism involving Keap1. Therefore, it regulated the downstream production of antioxidant enzymes, including NQO1 and HO-1. Overall, this finding suggested that TBSE could restore the redox state of H_2_O_2_-resistant HepG2 cells, indicating TBSE protected cells from H_2_O_2_-induced oxidative stress significantly. Beneficial resistance and effects on redox balance were attributed to activation of Nrf2. Present work revealed the potential health benefits of TBS and provided a test basis for developing functional food of TBS.

## 1. Introduction

During regular cellular processes, reactive oxygen species (ROS) are produced, comprising superoxide radicals, hydroxyl radicals, hydrogen peroxide, singlet oxygen, and various other free radicals. Under typical conditions, they serve a crucial function in maintaining physiological balance [1]. However, under pathological conditions, the overproduction of ROS can trigger oxidative stress, causing damage to various cellular components and contributing to numerous metabolic disorders, including Alzheimer’s disease, cardiovascular diseases, type 2 diabetes and various types of cancer [2]. The oxidative stress-induced cellular damage is primarily associated with the production of hydrogen peroxide (H_2_O_2_), because H_2_O_2_ diffuses more stably to larger tissue regions. Previously, the toxicity of H_2_O_2_ on lung epithelial cells has been studied [3]. H_2_O_2_-induced oxidative damage is an injury model for studying the positive effect of antioxidants on cells. Clinical evidence showed that oxidative stress occurred in a variety of conditions, including diabetes, neurodegenerative disorders, aging, cardiovascular ailments, and cancer [4]. More and more attention has been given to preventing and treating oxidative stress diseases by using antioxidant defense to remove ROS. Nucleus factor E2-related factor-2 (Nrf2) was found by previous researchers to possess antioxidative functions, which were related to oxidative stress [5].

As an important transcription factor, Nrf2 can regulate cellular antioxidant defense against oxidative stress by controlling the production of various antioxidant and detoxifying enzymes [6,7]. In a normal state, Nrf2 can form a complex with Kelch-like ECH-associated inhibitor 1 (Keap1), resulting in the complex being marked for degradation by the proteasome. When subjected to stress stimulation, the conformation of the complex changes, activating Nrf2 and increasing the nuclear translocation. Then, Nrf2 attaches to antioxidant response elements (ARE) within cell-protective genes, including those for glutathione S-transferase-1 (GST-1), quinone oxidoreductase-1 (NQO1), and heme oxygenase-1 (HO-1). Antioxidants, regulated by Nrf2, were proven to protect against liver conditions linked to oxidative stress and shield cells against harmful damage [8]. Finding natural sources with high antioxidant capacities such as medical plants, vegetables, and even cereals to reduce oxidative stress by inducing the Nrf2 pathway is attracting more and more research interest. Many phytochemicals have long been believed to improve oxidative stress through the Nrf2 signaling pathway. Chlorogenic acid alleviates apoptosis of human keratinocytes induced by deoxynivalenol by stimulating the Nrf2/HO-1 pathway [9]. Caffeic acid activates Nrf2 by targeting Keap1, reducing oxidative damage and lipid accumulation in liver cells, thereby improving metabolic dysfunction-related fatty liver disease [10]. Ferulic acid can enhance the activity of GSH-Px, CAT, and γ-GCS. Additionally, it mitigates H_2_O_2_-induced oxidative stress in HepG2 cells by modulating the Nrf2 signaling pathway [11]. The antioxidant effect of syringic acid can be attributed to the inactivation of cytoplasmic Nrf2, which prevents its translocation to the nucleus. In the nucleus, it can bind to the ARE promoter to induce genes encoding Hmox1 and NQO1 [12]. Rutin administration facilitates the translocation of nuclear factor erythroid 2-related factor 2 (Nrf2) from the cytoplasm to the nucleus, thereby triggering the expression of antioxidant and detoxifying enzymes, including heme oxygenase-1 (HO-1) [13]. Quercetin can achieve antioxidant effects by increasing the expression of CAT and SOD genes and upregulating Hmox1 [14]. The stimulation of a suite of downstream antioxidant and detoxifying enzymes governed by the Keap1-Nrf2 axis could be one of the mechanisms by which catechins ameliorate cadmium-induced injury [15].

Buckwheat and its sprouts, containing amino acids, peptides, high unsaturated fatty acids, flavonoids, and other phenolic compounds, are receiving increasing attention due to their beneficial nutritional components [16]. The content of flavonoids and flavonol glycosides in buckwheat sprouts is higher than that in seeds [17]. In addition, the antioxidant capacity of tartary buckwheat sprouts consistently exceeds that of common buckwheat sprouts. Extracts from buckwheat and buckwheat sprouts exhibited diverse pharmacological properties, encompassing effects such as antioxidant, anti-obesity, anti-fatigue, anti-inflammatory, and anti-diabetic properties [18]. Due to antioxidant related health benefits, buckwheat and its related products are receiving increasing public attention. For example, the literature has reported that tartary buckwheat flavonoids activated Nrf2 and protected hepatic cells against high glucose-induced oxidative stress [19]. Sprouts are known as a beneficial compound for health whose major bioactive compound is rutin and other flavonoids [20,21]. Quercetin, as the aglycon of rutin, is one of the most typical representatives of dietary antioxidants. It usually exists in fruits and grains, but is also found in tea [22]. Under oxidative stress, the representative effects of quercetin on several cells such as HepG2, L02, and macrophages have been reported [23]. However, studies on sprouts polyphenols with respect to relationship of antioxidant activities and molecular mechanisms are still lacking. Therefore, using quercetin as a positive control, we evaluated the ethanol extracts for their antioxidant properties and the possible protective effects of tartary buckwheat sprouts (TBSE) by establishing a H_2_O_2_-induced oxidative stress model in vitro. With the aim to reveal the acting mechanism of the Keap1-Nrf2 pathway, the study provided a possible dietary approach for the prevention of chronic diseases, a basis for the further development of buckwheat sprout products as well as a more in-depth theoretical basis for the utilization of buckwheat resources.

## 2. Materials and Methods

### 2.1. Materials and Reagents

We procured tartary buckwheat seeds from Dingbian Saixue cereal and oil Co. Ltd. in Yulin, China. Quercetin, rutin, chlorogenic acid, syringic acid, ferulic acid, dimethyl sulfoxide (DMSO), 3-(4,5-Dimethylthiazol-2-yl)-2,5-diphenyltetrazoliumbromide (MTT), 2,7-Dichlorodihydrofluorescein diacetate (DCFH-DA), and 2,2′-azobis (2-amidinopropane) dihydrochloride (ABAP) were purchased from Sigma-Aldrich Co. Ltd. (St Louis, MO, USA). Hank’s Balanced Salt Solution (HBSS) was acquired from Gibco Life Technologies in Shanghai, China. Additionally, a mitochondrial membrane potential assay kit containing JC-1 and nuclear and cytoplasmic extraction reagent kits were obtained from Beyotime Institute of Biotechnology in Shanghai, China.

SOD, CAT, GST, and MDA assay kits were sourced from the Jiancheng Bioengineering Institute in Nanjing, China. The BCA protein assay kit was procured from Sangon Biotech Co., Ltd. in Shanghai, China. The BCA protein assay kit was acquired from Sangon Biotech Co., Ltd. in Shanghai, China. Primary antibodies against NQO1 and Nrf2 were supplied by Abcam in Xi’an, China. HO-1, Keap1, β-actin and Histone H3 antibodies were purchased from Cell Signaling Technology Company in Shanghai, China. Labeled with horseradish peroxidase (HRP), species-specific secondary antibodies were procured from Santa Cruz Biotechnology in Xi’an, China. ECL Luminescent Solution was obtained from Millipore in Xi’an, China. Other chemicals meet analytical grade standards.

### 2.2. Experimental Design

In the present study, the effect of polyphenol extracts from tartary buckwheat sprouts (TBSE) on the redox system of HepG2-cell-induced oxidative injury by hydrogen peroxide was investigated for evaluating the protective effect and mechanism of tartary buckwheat sprouts (TBS). First, tartary buckwheat was subjected to germination treatment and polyphenol extract was prepared. Then, the extract was pretreated on HepG2 cells for a certain period of time, followed by induction with H_2_O_2_ to establish an oxidative damage model. Finally, a series of indicators representing the activity of oxidoreductases in cells were measured, such as reactive oxygen species content, mitochondrial membrane potential, CAA value, SOD, and other enzyme activities, to determine whether the tartary buckwheat sprouts polyphenol extract can upregulate the redox system. On this basis, the antioxidant mechanisms of phase II enzymes HO-1 and NQO1, nuclear transcription factor Nrf2, and related element Keap1 were speculated to determine whether they exert antioxidant effects through the Nrf2 pathway.

### 2.3. Extraction of Tartary Buckwheat Sprouts and HPLC Analysis

At room temperature, buckwheat seeds (Black Abundance Number One) were washed and immersed in distilled water for 12 h. Seeds were incubated under conditions at 20 °C and 80% humidity in the dark for the first two days and then under light:dark (12:12 h) conditions for eight days as described by Ling et al. [17]. Three times at intervals of 6–8 h every day, distilled water was sprayed. The sprouts were collected after 10 days with regular sprinkling. Fresh sprouts were dried with hot air at 60 °C and freeze-dried to obtain the final powder. Polyphenol compounds were extracted by refluxing with ethanol–water (70:30, *v*/*v*) for 10 h, which was repeated three times. After freeze drying, extracts were milled and stored at −20 °C for subsequent chemical analysis. The calorimetric method, based on the Folin–Ciocalteu assay as outlined by Siddhuraju et al. [24], was employed to measure the total phenolic content in the solvent extracts. As described by Jia et al. [25], the total flavonoid content was determined. Phenolic components of the extracts were analyzed by using an HPLC system (U-3000), from Thermo Fisher Scientific (Waltham, MA, USA), equipped with an auto sampler injector, a UV detector, and a Diamonsil C18 column (250 mm × 4.6 mm, particle size 5 µm; Dikma Technologies Co., Ltd., Beijing, China). Different concentrations of external HPLC standards including protocatechuic acid, gallic acid, chlorogenic acid, p-hydroxybenzoic acid, caffeic acid, syringic acid, rutin, coumaric acid, ferulic acid, and quercetin were prepared. The column temperature and UV detector were set at 30 °C and 280 nm, respectively. The injection volume was 20 µL, and the flow rate was 1.0 mL min^−1^. The gradient mobile phase consisted of acetonitrile and formic acid. The mobile phase A is made up of a mixture with a volume ratio of 0.1% formic acid, 5% acetonitrile, and the remainder being water at 94.9%; whereas mobile phase B consists of acetonitrile with 0.1% formic acid. The gradient program was set as follows: solvent A was changed linearly from 95% to 87% (0~5 min), changed linearly to 70% from 5 min to 25 min, and was then maintained at 70% for 10 min; then, solvent A was returned to 95% from 35 min to 45 min and maintained at 95% for the last 5 min. The quantity of each component is determined using the standard curve and the area under the peak. The levels of all phenolic compounds are presented in terms of milligrams per gram of dry weight (mg·g^−1^ DW).

### 2.4. Cell Culture and Treatments

Human hepatoma cells, the HepG2 cell line, originated from the Fourth Military Medical University in Xi’an, China. Fetal bovine serum and culture medium of RPMI 1640 were procured from Gibco Laboratories, Inc., in Xi’an, China. HepG2 cells were maintained in a humidified incubator at 37 °C and 5% CO_2_, using RPMI 1640 medium supplemented with 10% fetal bovine serum (FBS), 100 IU mL^−1^ penicillin, and 100 μg·mL^−1^ streptomycin.

Referring to a previous report [19], TBSE and quercetin were dissolved in DMSO to make a stock solution of 50 mg mL^−1^ and 40 mM, respectively, and then further diluted to final concentrations of 5, 10, 15, 20, 25 μg mL^−1^ for TBSE, and 1.25, 2.5, 5, 10, 20 μM for quercetin with a serum-free culture medium. In the culture medium, the final concentration of DMSO did not exceed 0.05%. The cells in the culture were first treated with a range of concentrations of TBSE or quercetin, followed by exposure to hydrogen peroxide for a set period of hours. The control group comprised cells that were not subjected to H_2_O_2_, whereas the model group included cells that were treated with H_2_O_2_. The cells used in the experiment were passaged for 15–18 generations and the culture medium was replaced daily. After processing, cells were collected for further analysis.

### 2.5. Assessment of Cell Viability

The MTT assay, a widely accepted approach for assessing cell viability, was employed to measure the reduction of tetrazolium salts as an indicator of cell survival. According to the method described by Chen et al. [26], cells were seeded at a concentration of 5 × 10^4^ cells per well in a 96-well plate containing 100 μL of growth medium and incubated at 37 °C for 24 h. The growth medium was then aspirated, and the cells were rinsed with PBS. A series of concentrations of ethanol extracts, quercetin or H_2_O_2_ treatments in 100 μL of 1640 RPMI, were added to the cells. Subsequently, the plates were placed in an incubator at 37 °C for 24 h. The treatment medium was subsequently removed, the cells were washed with PBS, and 10 μL of MTT solution (5 mg·mL^−1^) was added, followed by a further incubation at 37 °C for 4 h. After removing the culture medium, the purple formazan crystals that had formed were dissolved in 150 μL of dimethyl sulfoxide (DMSO) per well, with gentle shaking for 10 min. Then they were read in a microplate reader at 490 nm. In the test of protection from TBSE and quercetin, H_2_O_2_ (400 μM) was treated for 4 h after 24 h of TBSE (20 μg·mL^−1^) or quercetin (10 μM) pretreatment, and then operated as described above.

### 2.6. Measurement of Reactive Oxygen Species (ROS) and Assessing Mitochondrial Membrane Potential

Fluorescence marker DCFH-DA used to monitor intracellular ROS accumulation in HepG2 cells. HepG2 cells (1 × 10^5^ cells/well) were placed into laser confocal dishes, referring Chou’s method with slight modifications [23]. Cells in blank control group and the H_2_O_2_-induced group were incubated for 24 h with the medium. Cells assigned to the pretreatment group were cultured with TBSE at a concentration of 20 μg·mL^−1^ or quercetin at 10 μM for a period of 24 h. Then the induction of oxidative stress was performed by adding H_2_O_2_ (400 μM) to the culture medium for 4 h, except the blank control group incubated with RPMI 1640 for 4 h. Following incubation, the culture medium was aspirated, and the cells were rinsed twice using PBS. Then, DCFH-DA (10 μM) was mixed with 500 μL of RPMI 1640 medium and introduced to the confocal plate. After incubation for 30 min, the generation of ROS was determined by monitoring the fluorescence intensity at 535 nm, following excitation at 485 nm, and imaged by a confocal laser scanning microscope. The fluorescent indicator JC-1 demonstrates a voltage-dependent aggregation in the mitochondria, which is reflected by a shift in its emission spectrum from green (when in the monomeric state, signifying low MMP or depolarized mitochondria) to red (when in the aggregated state, indicating high MMP or polarized mitochondria). The group settings and operation of MMP test are the same as ROS test. After washing the cells twice with PBS, they were treated with the JC-1 staining reagent at 37 °C for 20 min. Then, they were rinsed three times with JC-1 staining buffer and then examined using a confocal microscope from Zeiss (Oberkochen, Germany).

### 2.7. Assessment of Cellular Antioxidant Activity (CAA)

The determination of Cellular Antioxidant Activity (CAA) was conducted following the method of Wolfe and Liu [27], with minor adjustments. HepG2 cells were plated at a concentration of 6 × 10^4^ cells per well in a 96-well microplate containing 100 μL of growth medium per well. Cells were grown in a CO_2_ incubator at 37 °C and 5% CO_2_ for 24 h. After removal of the growth medium, the wells were rinsed with 100 μL of PBS. HepG2 cells were then treated with 100 μL of varying concentrations of solutions (TBSE at 0, 5, 10, 20 μg·mL^−1^ or quercetin at 0, 2.5, 5, 10 μM) in an antioxidant treatment medium containing 25 μM DCFH-DA, for a duration of one hour. Following this, the medium was removed, and the wells were washed with 100 μL of PBS. Subsequently, 100 μL of ABAP solution (600 μM) in the oxidant treatment medium (HBSS with 10 mM Hepes) was added to each well. The 96-well plate was subsequently placed into a Fluoroskan Ascent microplate reader set at 37 °C, and fluorescence was recorded at 538 nm for excitation and 485 nm for emission at 5 min intervals for 60 min. Each plate also included wells for controls and blanks; the control wells had cells treated with DCFH-DA and oxidants, whereas the blank wells had cells treated with DCFH-DA in HBSS lacking oxidants. Quercetin was utilized as a reference standard to validate the experiment’s precision. After accounting for the blank values, the integrated area under the fluorescence over time curve was used to compute the CAA value of TBSE using the following formula.

CAA unit = 100 − (∫SA/∫CA) × 100. ∫SA refers to the cumulative area beneath the curve of sample fluorescence intensity over time, while ∫CA denotes the corresponding area for the control group. The median effect diagram, which depicts log (fa/fu) in relation to log (dose), is utilized to ascertain the median effective dose (EC50) of the plant-derived chemical extracts, where fa indicates the proportion of cells influenced by the treatment and fu indicates the proportion that remains uninfluenced. The EC50 value is then used to calculate the definitive CAA value. The antioxidant potency of isolated compounds is quantified as micromoles of quercetin equivalent (QE) per 100 μM of the compound, whereas the potency of the extracts is expressed as micromoles of QE per 100 g of material.

### 2.8. Assessment of MDA, SOD, CAT and GST Activities

The activity levels of the antioxidant enzymes were determined by following the protocol outlined in the assay kit’s manual. Following various treatments, cells were rinsed twice with PBS, then suspended in 1 mL of 0.1 M phosphate-buffered saline (pH 7.4) and processed through homogenization. Then centrifuge the homogenate at 3000× *g* for 15 min at 4 °C. The supernatant was collected for the determination of MDA, SOD, CAT, and GSH Px.

### 2.9. Preparation of Total and Nuclear Cell Lysates

Referring to the method reported by Chow et al. [23], total cell and nuclear cell lysates were prepared. First, HepG2 cells were plated at a concentration of 1 × 10^6^ cells per mL and incubated at 37 °C in a 5% CO_2_ environment for 24 h and then cells in the blank control group and H_2_O_2_-induced group were incubated for 24 h with medium. Cells in the pretreatment cohort were given TBSE at a dosage of 20 μg·mL^−1^ or quercetin at a concentration of 10 μM for a period of 24 h. Induction of oxidative stress is performed by adding H_2_O_2_ (400 μM) to the culture medium for 4 h except in the blank control group which is incubated with RPMI 1640 for 4 h. After being rinsed twice with PBS (pH 7.4), cells were lysed using a mammalian protein extraction reagent, along with kits designed for the extraction of nuclear and cytoplasmic components. Cells lysed with lysis buffer for 10 min contained 1% PMSF and 20 mM NaF in mammalian protein extraction kit on ice to detect NQO1, HO-1, Nrf2 and Keap1. To analyze the expression of cytoplasmic Nrf2, Keap1 and nuclear Nrf2, a nuclear extraction reagent was used for cells lysing. They were gently scraped with a cell scraper, and the dissolved cells were removed from the culture dish and transferred to a microcentrifuge tube. Samples were subjected to centrifugation at 4 °C for 20 min at a speed of 12,000× *g*. The resulting supernatant was gathered, aliquoted evenly, and kept at −80 °C for subsequent Western blot analysis.

### 2.10. Western Blotting Analysis

According to the method of Shi et al. [6], total protein content was determined and isolated using a BCA protein assay kit. Equal volumes of protein samples, each 30 μg, were resolved by 10% SDS-PAGE and then transferred to a membrane. The membrane was initially incubated in a blocking solution containing 1% BSA, followed by an overnight incubation at 4 °C with the primary antibodies: anti-HO1 (1:1000), anti-NQO1 (1:10,000), anti-Keap1 (1:1000), anti-Nrf2 (dilution 1:1000), anti-β-actin (1:5000), and anti-Histone H3 (1:2000). Afterward, the membrane was washed with TBST and further incubated for one hour at room temperature with peroxidase-labeled secondary antibodies (dilution 1:10,000). Blots were visualized using an enhanced chemiluminescence (ECL) detection reagent provided by Pioneer Technology Ltd. in Xi’an, China. The densitometric analysis of the bands was conducted with a UVP Bioimaging System from Upland, CA, USA.

### 2.11. Statistical Analysis

Results were displayed as average ± standard deviation (*n* = 3). A one-way ANOVA test was utilized for statistical analysis. If the analysis of variance showed statistical significance, then Duncan’s correction was performed. There were significant differences in the column values that were superscripted with different letters (*p* < 0.05).

## 3. Results

### 3.1. Phenolic Compounds of Ethanol Extracts of Tartary Buckwheat Sprouts

The determination results of total phenols and flavonoids were as shown in Table 1. It showed the content of total phenol was 271.21 ± 8.41 mg·g^−1^ DW, and the quantity of total flavones was measured at 367.31 ± 9.25 mg·g^−1^ dry weight (DW). In the ethanolic extracts of tartary buckwheat sprouts, phenolic compounds were detected by comparing the retention time with polyphenol standards using HPLC, and their corresponding chromatograms are depicted in Figure 1. Six predominant constituents—chlorogenic acid, caffeic acid, syringic acid, rutin, ferulic acid, and quercetin—were proposed to be in the TBSE by comparison of retention time with previous HPLC analysis, as illustrated in Figure 1B. Rutin was the richest phenolic component in TBSE as well as its content reached 105.15 ± 34.39 mg·g^−1^ DW. There was almost no significant difference in the contents of chlorogenic acid, ferulic acid and quercetin and contents were less than 5 mg·g^−1^ DW, respectively. Although the content is low, quercetin has highly bioactive and quinone reductase-inducing activity at the cell level [28]. Caffeic acid and syringic acid are the least ingredients among these components. Our research on content of phenolic compounds is similar to previous reports [16,17,18]. The observations demonstrated that TBSE had higher total flavonoid contents and polyphenol components.

### 3.2. Cell Survival and Protection from Oxidative Stress

To investigate the effects of TBSE, Que, and H_2_O_2_ on cell survival, HepG2 cells were, in different concentrations of solvents, incubated for 24 h and then tested by MTT assay. Firstly, we determined cell survival in the presence of TBSE alone at different concentrations of 5 μg·mL^−1^, 10 μg·mL^−1^, 15 μg·mL^−1^, 20 μg·mL^−1^ and 25 μg·mL^−1^ for 24 h. As shown in Figure 2A, when the concentration of TBSE reached 20 μg·mL^−1^, there was no significant fluctuation in cell viability. The cell viability was stable or slightly higher than 100%, indicating that TBSE did not cause cell damage with the present concentration range. So, it was suitable for further research with these concentrations. While the concentration reached 25 μg·mL^−1^, the cell viability decreased slightly. Cell survival in the presence of quercetin alone at these concentrations (1.25 μM, 2.5 μM, 5 μM, 10 μM and 20 μM) for 24 h was determined. Results from Figure 2B showed the concentration of quercetin was suitable for further investigation within 10 μM. When the concentration of quercetin reached 20 μM, the cell viability reduced and exhibited a slight decline. Subsequently, the MTT assay of H_2_O_2_-induced oxidative injury on HepG2 was tested. Previous research suggested that exposure to H_2_O_2_ may lead to cytotoxicity in HepG2 cells [29]. So, we used H_2_O_2_ to establish an oxidative damage model and it demonstrated that cell viability decreased while H_2_O_2_ concentration increased. In addition, 400 μM was the appropriate concentration of H_2_O_2_ for the next experiments in which the cell viability was approximately at 50% (Figure 2C).

To investigate the ability to protect the cells against oxidative damage, HepG2 cells were treated with TBSE or quercetin for 24 h prior to H_2_O_2_ exposure for 4 h. The MTT assay indicated that the cytotoxicity of TBSE was notably diminished in a dosage-dependent manner (*p* < 0.05), as well as with quercetin treatment, when compared to the group treated exclusively with H_2_O_2_ (as shown in Figure 2D,E). These results revealed that TBSE and quercetin could resist H_2_O_2_-induced cytotoxicity at appropriate concentrations, less than 20 μg·mL^−1^ and 10 μM, respectively. As the concentration increased, the protective ability was also enhanced. Thus, the best protective effect reached in treatment of 20 μg·mL^−1^ of TBSE and 10 μM of quercetin, which were, respectively, also suitable concentrations in the following tests.

### 3.3. Effect of TBSE on H_2_O_2_-Triggered ROS Generation and Alterations in Mitochondrial Membrane Potential

As is well known, the overproduction of ROS may result in cellular redox disorder. Experiments have shown that supplementing sources rich in antioxidants can significantly regulate cellular redox by reducing ROS [30]. Therefore, we examined the effect of TBSE and quercetin on H_2_O_2_-induced ROS generation with DCFH-DA by laser confocal microscopy. Compared to the untreated control group (Figure 3A,C), the results indicated that exposure to H_2_O_2_ could lead to a rapid and significant increase of green fluorescence intensity in the cellular ROS levels (*p* < 0.05). TBSE and quercetin pretreatment significantly reduced ROS levels, and TBSE showed better performance than quercetin in clearing intracellular ROS (*p* < 0.05). The inhibition of excessive ROS production induced by H_2_O_2_ may be partially attributed to various phenolic compounds in TBSE.

In mitochondrial membrane potential (MMP), excessive ROS production leads to a decreased effect, thus inducing apoptotic cell death [31]. This phenomenon may be caused by mitochondrial dysfunction. Therefore, we assessed if an exposed cell under H_2_O_2_ could cause MMP decrease using JC-1 fluorescence probe. Compared with the H_2_O_2_ treatment alone (Figure 3B,D), JC-1 fluorescence intensity showed that TBSE and quercetin could balance mitochondrial membrane potential disruption induced by H_2_O_2_. The green fluorescence significantly increased compared with the control group. Meanwhile, it showed that MMP declined after treatment with H_2_O_2_ for 4 h (*p* < 0.05). Afterwards, we assessed the protective influence of TBSE and quercetin against the H_2_O_2_-triggered decrease in MMP. In the presence of TBSE, the green fluorescence decreased and red fluorescence increased significantly compared to the group treated by H_2_O_2_ alone (*p* < 0.05). Similarly, the decreased effect of quercetin against MMP, induced by H_2_O_2_, exhibited the same trend while the protective effect was lower than TBSE. In contrast to the control cells, the treatment of HepG2 cells with H_2_O_2_ led to a diminished ratio of red to green fluorescence, as observed under confocal microscopy, which was restored by treatment with TBSE or quercetin. Therefore, our research results indicate that both TBSE and quercetin can significantly inhibit ROS related mitochondrial dysfunction, and TBSE has a better protective effect.

### 3.4. Cellular Antioxidant Activity Assay and Measurement of MDA, SOD, CAT, and GST Activities

To evaluate the antioxidant quality of TBSE exactly, we measured the cellular antioxidant activity (CAA) of TBSE. CAA serves as a quantitative metric for the antioxidant potency of phytochemicals and dietary supplements, expressible in micromoles of quercetin equivalent (QE) per 100 g of fresh weight (FW). Using this assay, we found TBSE had a higher value in cellular antioxidant activity, whose content is about thirty times than that of black tea extracts and Epigallocatechin gallate [32]. As shown in Figure 4A, the CAA value of TBSE exceeded that of quercetin, signifying the significant antioxidant function of TBSE in combating ABAP-generated peroxyl radicals within HepG2 cells, which is achieved by inhibiting the oxidation of DCFH and the subsequent formation of DCF. The results indicated that the TBSE showed more significant protective effect than quercetin in promoting the antioxidant capacity of cells. To examine the protection of TBSE on H_2_O_2_-induced oxidation stress, we speculated whether the protective effect of TBSE is related to MDA levels and enzyme activities. In Figure 4B, exposing HepG2 cells to H_2_O_2_ significantly increased intracellular MDA levels (*p* < 0.05). When HepG2 cells were incubated with TBSE or quercetin, it significantly inhibited the H_2_O_2_-induced increase in MDA levels (*p* < 0.05). On the contrary, in Figure 4C–E, it was observed that there was a notably reduced activity of SOD, CAT, and GST in the groups treated with H_2_O_2_ when compared to the control groups (*p* < 0.05). Pretreatment with TBSE or quercetin markedly inhibited the reduction in SOD activities, CAT activities, and GST activities (*p* < 0.05).

### 3.5. Expression of NQO1, HO-1, Keap1, and Nuclear Transfer of Nrf2

More and more evidence suggests that the Keap1/Nrf2 antioxidant defense pathway serves as a crucial regulatory element in the cells’ adaptive response to oxidative stress, mediating a series of antioxidant response genes and phase II detoxifying enzymes, including HO-1 and NQO1. As widely recognized, NQO1 and HO-1 are among the primary antioxidant enzymes that are vital in the antioxidant defense mechanisms triggered by H_2_O_2_ in hepatic cells [33]. Therefore, to investigate whether TBSE leads to the expression of antioxidant proteins, the protein expressions of NQO1, HO-1 and total Nrf2 were measured. Compared with the H_2_O_2_ treatment alone, TBSE and quercetin could attenuate H_2_O_2_-induced reduction of antioxidant proteins expression. Figure 5A, B indicated that the treatment of cells with TBSE and quercetin markedly increased the expression of NQO1, HO-1, and total Nrf2 than with H_2_O_2_ treatment alone (*p* < 0.05). Furthermore, we identified the nuclear translocation status of Nrf2. Cells were pretreated with TBSE and quercetin before treated with H_2_O_2_. Then, the cytoplasm and nucleus were separated from each other and evaluated. As depicted in Figure 5A,C, in contrast to the untreated control group and the cells treated with H_2_O_2_, nuclear Nrf2 protein in the pretreatment group showed an increased state while cytoplasmic Nrf2 decreased (*p* < 0.05). Moreover, the expression of relevant factors induced by TBSE was more obvious than quercetin induced. While the trend of cytoplasmic Keap1 was the opposite of cytoplasmic Nrf2, the expression of nuclear Keap1 was negligible in each group. Perhaps it is in connection to the fact that Keap1 is a negative feedback regulator of Nrf2. The Nrf2 of the treated group obviously transformed from cytoplasm to nucleus. We suspected that the antioxidant ability of the TBSE and quercetin have a relationship with the nucleus translocation of Nrf2. This is consistent with the relevant literature [29]. This result clearly indicated that the cytoprotective effects of TBSE were monitored through translocation and up-regulation of Nrf2.

## 4. Discussions

Oxidative stress emerges as an abnormal condition in the human body when the generation of free radicals outpaces the antioxidant defense system. The expression of free radicals and other ROS will damage the basic macromolecules of cells, leading to the development of many human diseases, such as inflammation, atherosclerosis, rheumatoid arthritis, neurodegenerative diseases, and cancers [34]. Numerous investigations have demonstrated that tartary buckwheat flavonoids exert favorable antioxidant actions in both animals and humans, with key constituents like rutin and quercetin positively impacting oxidative stress, suggesting their potential as a therapeutic candidate for disease prevention [35]. However, a clear molecular mechanism underlying these effects has not been established. In the ethanol extracts of tartary buckwheat sprouts (TBSE) can effectively prevent the imbalance of redox status in HepG2 cells by reducing the generation of ROS and modulating the functionality of antioxidant enzymes. As normal live cells in human (L02), HepG2 cells possess the same biological activity, and the liver is the main source of energy expenditure [27]. H_2_O_2_, inducing the self-generation of free radicals, is called ROS release [36]. Our results showed that treating HepG2 cells with H_2_O_2_ led to a decrease in cell survival rate, an increasing in ROS production, and mitochondrial dysfunction. Excessive production of ROS can damage mitochondrial membranes and emit mitochondrial apoptotic factors into the cytoplasm, subsequently triggering caspase activation, ultimately leading to cell apoptosis [37]. Natural antioxidants that curb the generation of ROS are deemed to play a crucial role in safeguarding the liver against oxidative damage. Therefore, previous investigation has mainly focused on natural antioxidants with chemical preventive effects and their mechanisms of action [38]. However, up until now, there have been no studies on the effect of TBSE on oxidative stress in liver cells triggered by H_2_O_2_. In this study, we investigated the protective impact of TBSE against H_2_O_2_-induced harm in HepG2 cells. The findings indicated that subjecting HepG2 cells to H_2_O_2_ considerably lowered the cell survival rate, suggesting that TBSE could shield HepG2 cells from H_2_O_2_-induced damage.

Plant phenolic substances provide parallel protection by regulating the antioxidant system [39]. Phenolic compounds of TBS have been widely studied and reported by using HPLC, however, there is still some controversy over the types of polyphenols. It was reported that rutin was the only major flavonoid constituent of TBS, whereas other polyphenols presented only in trace amounts or were not detected [40]. However, Kim et al. found that chlorogenic acid, rutin, quercetin, and four C-glycosylflavones (orientin, isoorientin vitexin, isovitexin) appeared in the edible parts of TBS, and chlorogenic acid content was much higher than that of any other flavonoid, except rutin [16], which is similar to present result. Six phenolic compounds (caffeic acid, syringic acid, rutin, ferulic acid, and quercetin) were detected in the TBSE in this investigation. The main reason for this difference might be due to the diverse tartary buckwheat varieties and different extraction and detection methods of polyphenols. In terms of detection methods, although HPLC is a mature and widely used method for phenolic compounds analysis of mixed extracts, it still has some shortcomings. The separation efficiency is not high enough and there is a certain degree of deficiency in accuracy. Standard compound co-injection analysis can make up for this deficiency and has been widely used in polyphenols analysis [41], which is also a method worth learning in subsequent research. It can improve the accuracy and sensitivity by eliminating variations in chromatographic systems and enhancing the signal intensity of target polyphenols, particularly important in low concentration sample with polyphenols presenting only in trace amounts. Encouragingly, however, six phenolic compounds detected in the TBSE, particularly rutin and quercetin, have been reported to improve oxidative stress. According to reports, both rutin and quercetin attract favorable changes of antioxidant system in HepG2 cells, which may delay cellular oxidative stress [42]. Quercetin offers protection against oxidative stress in human hepatocellular carcinoma cell lines [43]. What’s more, quercetin has stronger quinone reductase-inducing activity at the cell level, because it has good stability and lower molecular weight, which makes it easier to enter the cell membrane. Meanwhile, evidence suggest that rutin has a significant scavenging effect on oxygen free radicals and plays a role in regulating oxidative stress via preventing the appearance of ROS while it is less stable than quercetin in vivo [44,45,46].

It was demonstrated that TBSE was able to decrease free radical production and improve mitochondrial function through significantly attenuating HepG2 cells injury induced by H_2_O_2_ [47]. The mitochondria serve as the primary source of ROS in a majority of mammalian cells. Redundant ROS led to mitochondrial injury in various pathological conditions [48]. Damaged mitochondria can affect lipid homeostasis, bioenergetics, and ROS increasing, which accumulate lipid content and oxidative stress [49]. More and more evidence suggests that MMP can act as a more precise diagnostic approach for detecting early mitochondrial dysfunction [50]. In this study, we used JC-1 to reveal the loss of MMP induced by H_2_O_2_ in HepG2 cells. After 24 h of TBSE pretreatment, MMP returned to baseline levels. These results suggested that the protection of TBSE to alleviate oxidative stress that contributed to the inhibition of mitochondrial apoptosis. The CAA assay is a potent biological technique that captures the cellular uptake, metabolic transformation, and distribution of antioxidants. The measurement principle of CAA is slightly different from the determination of ROS. Initially, the non-polar DCFH-DA is taken up by HepG2 cells and subsequently deacetylated to DCFH by cellular esterase. Concurrently, the peroxide radicals generated from ABAP breakdown can assault the cell membrane, leading to the production of additional free radicals and the oxidation of DCFH to DCF [27]. Simply, the fluorescence level formed by HepG2 cells is related to the oxidation level [51]. By measuring the fluorescence value, the CAA value was finally converted and calculated. Moreover, the assay showed that TBSE had higher cellular antioxidant activity compared to quercetin which indicated TBSE had a strong potential to induce intracellular antioxidant activity.

Oxidative stress could induce cell injury, which is related to production of MDA and ROS, such as H_2_O_2_, hydroxyl radicals and peroxynitrites [52]. Thus, the level of MDA within cells serves as an additional marker that is believed to be a primary cause of oxidative processes. In our research, we have shown that subjecting liver cells to H_2_O_2_ leading to an excessive generation of reactive oxygen species (ROS), thereby causing oxidative stress. However, treatment with TBSE markedly suppressed the ROS generation induced by H_2_O_2_, potentially owing to TBSE’s potent antioxidant and radical-scavenging capabilities. SOD acts as an eliminator of oxygen radicals by transforming superoxide anions into H_2_O_2_, which is subsequently degraded into water by CAT and GST [53]. In our results, the treatment of HepG2 cells with H_2_O_2_ resulted in markedly reduced levels of SOD, CAT, and GST, but a 24 h pre-treatment with TBSE was observed to counteract this effect.

Nrf2, a basic leucine zipper (bZIP) transcription factor and a component of the cap’n’collar (CNC) family, plays a pivotal role in modulating the expression of phase II enzymes at the transcriptional level. The Nrf2 activation pathway acts as a swift feedback mechanism that is triggered when cells are subjected to oxidative stress [54]. It is documented that both nuclear and total Nrf2 levels can be stimulated in response to H_2_O_2_ exposure, serving as an adaptive mechanism to shield cells against oxidative stress and injury [55], a finding that aligns with our observations. Additionally, it has been demonstrated that quercetin can stimulate the translocation of Nrf2 and enhance the antioxidant defense system. We speculated that the elevated protein levels of NQO1 and HO-1 were a result of the Nrf2 signaling pathway being triggered. As anticipated, TBSE was found to activate Nrf2, which in turn led to an elevation in the protein levels of NQO1 and HO-1. While our research indicated that TBSE could mitigate oxidative harm in vitro by triggering the Nrf2 pathway, it is important to consider whether other redox-sensitive transcription factors, such as NF-KB, might also regulate the expression of antioxidant enzymes. Furthermore, it remains to be determined whether TBSE could facilitate the movement of Nrf2 from the cytoplasm into the nucleus through alternative signaling pathways, such as the AMPK and GSK3β pathways.

## 5. Conclusions

The present study identified several major phenolic compounds of ethanol extracts from tartary buckwheat (TBSE), namely chlorogenic acid (3.38 ± 1.62 mg·g^−1^ DW), caffeic acid (0.23 ± 0.18 mg·g^−1^ DW), syringic acid (0.42 ± 0.27 mg·g^−1^ DW), rutin (105.15 ± 34.39 mg·g^−1^ DW), ferulic acid (1.52 ± 0.50 mg·g^−1^ DW), and quercetin (2.67 ± 2.73 mg·g^−1^ DW). The investigation established that TBSE offers protection against cell toxicity caused by H_2_O_2_. Our studies revealed that TBSE is effective in lowering the formation of reactive oxygen species (ROS), preventing the decline of mitochondrial membrane potential (MMP), and increasing the action of superoxide dismutase (SOD), catalase (CAT), and glutathione S-transferase (GST) enzymes. It is mainly manifested in HepG2 cells exposed to a range of TBSE concentrations (0, 5, 10, 15, 20, and 25 μg mL^−1^) for a duration, as well as exposed to a range of concentrations of quercetin (0, 1.25, 2.5, 5, 10 and 20 μM), respectively. In which, the antioxidant effects of 20 μg mL^−1^ TBSE and 10 μM Que on H_2_O_2_ induced oxidative damage in HepG2 cells were the most significant. This is reflected in its ability to minimize ROS content and mitochondrial membrane potential changes to the greatest extent possible, with the lowest levels of malondialdehyde and the highest antioxidant enzyme activity of enzymes. In samples pretreated with 20 μg mL^−1^ TBSE and 10 μM Que, NQO1, HO-1, and Nrf2 showed upregulated expression in immunoblotting experiments.

This discovery suggests that the protective effect of TBSE is likely due to its interaction with the negative feedback regulation of Keap1 and the nuclear translocation of Nrf2. In summary, TBSE can significantly reduce oxidative damage caused by H_2_O_2_, potentially offering a means to prevent health issues associated with oxidative stress. This work provides a possible dietary approach for prevention of chronic diseases, a basis for the further development of buckwheat sprout functional foods as well as the in-depth utilization of buckwheat resources.

## Figures and Tables

**Figure 1 foods-13-03726-f001:**
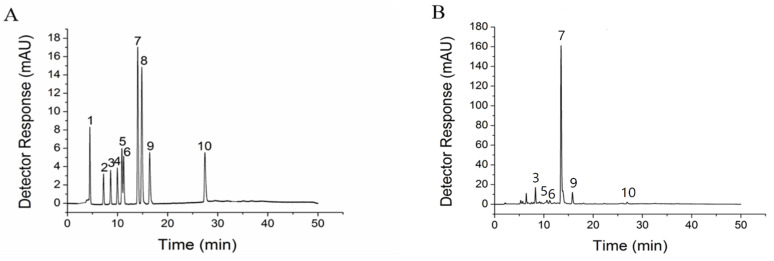
HPLC of the ethanol extracts from the tartary buckwheat sprouts at wavelength 280 nm. (**A**) Standards. Compounds numbered 1 through 10 correspond to the following: gallic acid, protocatechuic acid, chlorogenic acid, *p*-hydroxybenzoic acid, caffeic acid, syringic acid, rutin, coumaric acid, ferulic acid, and quercetin. (**B**) TBSE. Compounds 3, 5, 6, 7, 9 and 10 respectively represent chlorogenic acid, caffeic acid, syringic acid, rutin, ferulic acid, and quercetin.

**Figure 2 foods-13-03726-f002:**
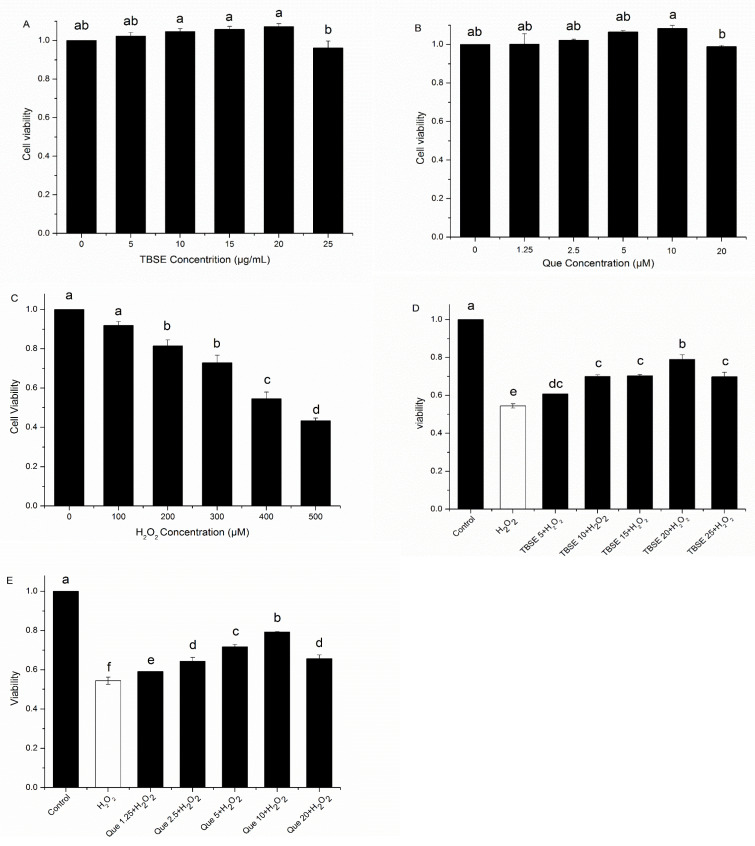
Effect of cell viability treated by TBSE and quercetin and H_2_O_2_-induced cytotoxicity in HepG2 cells. (**A**) The survival rate of HepG2 cells exposed to a range of TBSE concentrations (0 μg·mL^−1^, 5 μg·mL^−1^, 10 μg·mL^−1^, 15 μg·mL^−1^, 20 μg·mL^−1^, and 25 μg·mL^−1^) for a duration of 24 h. (**B**) The survival rate of HepG2 cells exposed to a range of concentrations of quercetin (0 μM, 1.25 μM, 2.5 μM, 5 μM, 10 μM and 20 μM) for 24 h respectively. (**C**) The survival rates of HepG2 cells exposed to a variety of H_2_O_2_ concentrations, from 0 to 500 μM, for a period of 24 h each. (**D**) Survival rates of HepG2 cells that were pre-treated with various concentrations of TBSE for 24 h, followed by 4-h incubation with 400 μM H_2_O_2_. (**E**) HepG2 cells were first treated with various concentrations of quercetin for the duration of 24 h, followed by exposure to 400 μM H_2_O_2_ for an additional 4 h. Each value was depicted as the average ± standard deviation (*n* = 3). Entries in a column with different letters denote statistically significant differences (*p* < 0.05).

**Figure 3 foods-13-03726-f003:**
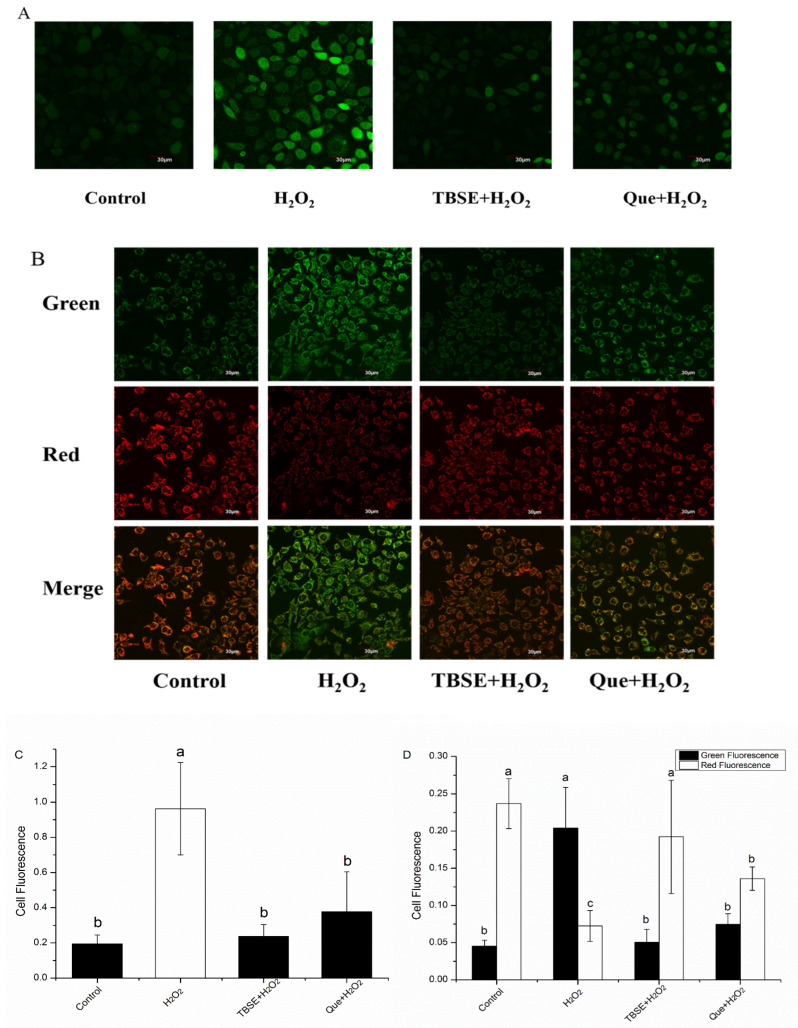
Influence of TBSE and quercetin on ROS generation and MMP decline caused by H_2_O_2_ in HepG2 cells. (**A**) Effect of TBSE and quercetin on H_2_O_2_-induced ROS production in HepG2 cells, through treatment with 20 μg·mL^−1^ TBSE or 10 μM quercetin for 24 h, treatment with 400 μM H_2_O_2_ for 4 h, then incubated with 10 μM of DCFH-DA at 37 °C for 30 min, cells were washed with PBS and captured by confocal microscope. (**B**) Effect of TBSE and quercetin against H_2_O_2_-induced mitochondrial membrane potential (MMP) after treatment with 20 μg·mL^−1^ TBSE or 10 μM quercetin for 24 h, treatment with 400 μM H_2_O_2_ for 4 h, HepG2 cells were treated with 10 μM JC-1 and incubated for 30 min at a temperature of 37 °C and then washed with PBS and captured by confocal microscope. (**C**) Measurement of the relative fluorescence intensity of ROS. (**D**) Measurement of the relative fluorescence intensity indicative of MMP. Each value was depicted as the average ± standard deviation (*n* = 3). Entries in a column with different letters denote statistically significant differences (*p* < 0.05).

**Figure 4 foods-13-03726-f004:**
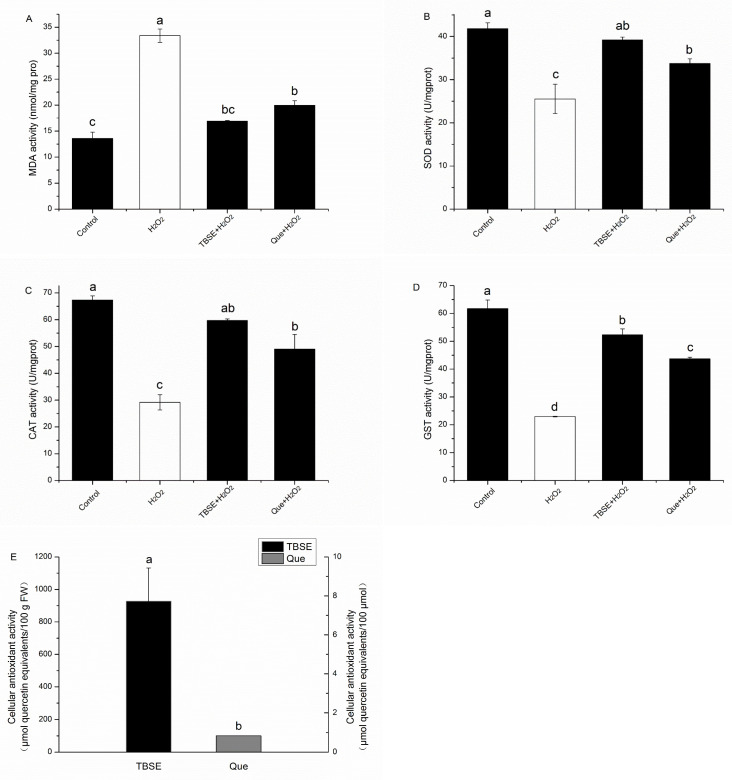
Effect of TBSE and quercetin on MDA, CAT, SOD, GST, and cellular antioxidant activity (CAA) in HepG2 cells. Incubating with TBSE (20 μg·mL^−1^) or quercetin (10 μM) for 24 h, cells then treatment with H_2_O_2_ (400 μM) for 4 h for determination of antioxidant enzymes. (**A**) MDA level. (**B**) SOD enzyme activity. (**C**) CAT enzyme activity. (**D**) GST enzyme activity. (**E**) Cell antioxidant capacity. Each value was depicted as the average ± standard deviation (*n* = 3). Entries in a column with different letters denote statistically significant differences (*p* < 0.05).

**Figure 5 foods-13-03726-f005:**
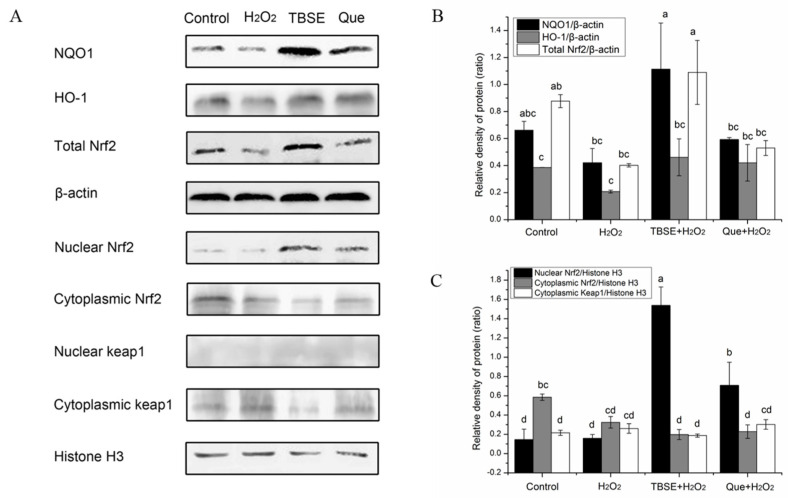
TBSE and quercetin restored H_2_O_2_-induced impairment of the Keap1/Nrf2 pathway in HepG2 cells. Incubating with TBSE (20 μg·mL^−1^) or quercetin (10 μM) for 24 h prior to treatment with H_2_O_2_ (400 μM) for 4 h. Cells lysates were processed and Western blot assay was performed to track the expression levels of specific proteins. (**A**) Expressions of NQO1, HO-1, total Nrf2, nuclear Nrf2, cytoplasmic Nrf2, nuclear Keap1, and cytoplasmic Keap1, were detected by Western blotting. (**B**) Densitometric quantification of HO-1, NQO1, and total Nrf2 levels. β-actin served as a reference protein. (**C**) Densitometric quantification of cytoplasmic and nuclear Nrf2 as well as Keap1. Histone H3 was used as an internal control. Each value was depicted as the average ± standard deviation (*n* = 3). Entries in a column with different letters denote statistically significant differences (*p* < 0.05).

**Table 1 foods-13-03726-t001:** The total flavonoids contents and phenolic components of tartary buckwheat sprouts ethanol extracts (mg·g^−1^ DW) in vitro.

Contents	TFC	TPC	Chlorogenic Acid	Caffeic Acid	Syringic Acid	Rutin	Ferulic Acid	Quercetin
TBSE	367.31 ± 9.25	271.21 ± 8.41	3.38 ± 1.62	0.23 ± 0.18	0.42 ± 0.27	105.15 ± 34.39	1.91 ± 0.06	0.83 ± 0.13

TBSE indicates tartary buckwheat sprouts ethanol extracts. TFC and TPC indicate total flavonoids contents and total polyphenols contents. Each value was depicted as the average ± standard deviation (*n* = 3). Entries in a column with different letters denote statistically significant differences (*p* < 0.05).

## Data Availability

The original contributions presented in this study are included in this article. Further inquiries can be directed to the corresponding author.

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
