# Peer review of "Extracts from Tartary Buckwheat Sprouts Restricts Oxidative Injury Induced by Hydrogen Peroxide in HepG2 by Upregulating the Redox System"

_foods, 2024, doi:10.3390/foods13233726_

Round 1

Reviewer 1 Report

Comments and Suggestions for Authors

After reading the article Extracts from tartary buckwheat sprouts restricts oxidative injury induced by hydrogen peroxide in HepG2 by up-regulating the redox-system, the results contained in the results are very interesting, however, I have several suggestions:

Minor

1. Put the reference number in brackets in the text.

2. Divide the discussion results.

3. Reference the techniques used in materials and methods.

4. In the references section, attach the DOI, the name of the journal in italics, and the year of publication in bold.

Major

1. In the materials and methods section, attach a section on experimental design where you explain what was done.

2. How was TBSE dissolved and QUE?

3. Why was QUE used?

4. Discuss based on the compounds that were found with the results found

5. Be more specific in the conclusion.

Reviewer 2 Report

Comments and Suggestions for Authors

Li and co-workers present a study on the effect of polyphenol extracts from Tartary buckwheat sprouts on the redox system of HepG2 cell-induced oxidative injury by hydrogen peroxide. The main aim was to evaluate the protective effect and mechanism of Tartary buckwheat sprouts.

The authors reported the detection of six phenolic compounds (chlorogenic acid, syringic acid, caffeic acid, rutin, ferulic acid, and quercetin), which were associated with the antioxidant effects of the evaluated extract. In fact, the antioxidant activity of phenolic compounds biosynthesized by plants is well-reported in the specialized literature. The Introduction section should provide a deeper discussion about this topic, with results about the antioxidant mechanism of phenolic secondary metabolites.

The study should be interesting for the research area.

Section 2.2 must be completely revised.

Please, in the phrase: “Polyphenols compounds were produced by refluxing”, change to “extracted by refluxing”. “Phenolic components of the extracts were tested” by “were analyzed”

More details about the HPLC system should be provided, for instance, elution time, flow, column type, HPLC detector type

Section 3.3. The use of “identified compound” should be avoided as no compound was isolated and characterized by spectroscopic and spectrometric methods (NMR, IR, mass data). The authors should state that the phenolic compounds were detected in the extract and their characterization was based on previous analysis of standard and pure compounds at the same analytical method.

In addition, an additional co-injection analysis is important for the more consistent proposition of the chemical structures. Chlorogenic acid, Caffeic acid, Syringic acid, Rutin, Ferulic acid, and Quercetin were proposed to be in the extract by comparison of retention time with previous HPLC analysis. These six compounds could be added in the extract and a new HPLC analysis will reinforce the inference about the presence of them in the evaluated extract.

Round 2

Reviewer 1 Report

Comments and Suggestions for Authors

The authors responded to the suggestions

Author Response

The authors responded to the suggestions.

Response: We appreciate for your valuable approval. We will continue to take our efforts to provide the journal with more high-quality scientific research achievements.